# Projective Synchronization of Delayed Uncertain Coupled Memristive Neural Networks and Their Application

**DOI:** 10.3390/e25081241

**Published:** 2023-08-21

**Authors:** Zhen Han, Naipeng Chen, Xiaofeng Wei, Manman Yuan, Huijia Li

**Affiliations:** 1School of Cybersecurity, Northwestern Polytechnical University, Xi’an 710072, China; 2International School, Beijing University of Posts and Telecommunications, Beijing 100876, China; 3The 20th Research Institute of China Electronics Technology Group Corporation, Xi’an 710068, China; 4School of Computer Science, Inner Mongolia University, Hohhot 010021, China; 5School of Science, Beijing University of Posts and Telecommunications, Beijing 100876, China; hjli@amss.ac.cn

**Keywords:** nonlinear control mechanism, memristor, coupled neural networks, projective synchronization, secure communication

## Abstract

In this article, the authors analyzed the nonlinear effects of projective synchronization between coupled memristive neural networks (MNNs) and their applications. Since the complete signal transmission is difficult under parameter mismatch and different projective factors, the delays, which are time-varying, and uncertainties have been taken to realize the projective synchronization of MNNs with multi-links under the nonlinear control method. Through the extended comparison principle and a new approach to dealing with the mismatched parameters, sufficient criteria have been determined under different types of projective factors and the framework of the Lyapunov–Krasovskii functional (LKF) for projective convergence of the coupled MNNs. Instead of the classical treatment for secure communication, the concept of error of synchronization between the drive and response systems has been applied to solve the signal encryption/decryption problem. Finally, the simulations in numerical form have been demonstrated graphically to confirm the adaptability of the theoretical results.

## 1. Introduction

A fact that is well-known is that a chaotic system is essential in signal processing and secure communication. However, some existing results for secure communication are still based on traditional neural networks [1,2,3]. Actually, the memristor, the fourth basic element of electrical circuits, was firstly raised by Prof. Leon Chua [4] in 1971; the memristor exhibited better chaotic characteristics than the resistor in mimicking the synaptic plasticity. Depending on such excellent attributes in biologicals, many scholars have combined the memristor with neural networks (NNs) to propose the memristive neural networks (MNNs) [5,6,7,8,9] for a better understanding of the structure and functions of brain networks. Nevertheless, up to now, few researchers have conducted the memristor to the chaotic system for signal encryption and decryption, which inspired us to consider the MNNs for secure communication from the biological point of view.

In recent years, due to the superior mimic of the human brain, the synchronization of MMNs has attached considerable attention from various fields and became a hot topic. It is worth mentioning that as a principle dynamic behavior of MNNs, the accuracy of synchronization determines the validity of encryption and decryption. Therefore, the potential application of synchronization has extended to various areas, such as in image protection [10], social networks [11], pattern recognition, etc. In [12], the cluster output synchronization was investigated and two different control approaches were proposed. Finite-time and fixed-time synchronization were investigated in [13,14]. Considering the sensitivity of the channel for secure communication, Ref. [15] studied Dos attacks between the master and slave MNNs, and the results were applied to image encryption. However, the problem with faster communication still exists [16] regardless of whether extending binary digital to M-nary digital. For this issue, the projective synchronization was first introduced by Mainieri et al. [17].

In a practical circumstance, different synchronizations are necessary between the drive-response MNNs, even more so for secure communication. The impact caused by various projective factors for different structures of neural networks has been regarded in many studies [18,19,20]. Fu et al. [18], under a pinning control scheme, studied the projective synchronization for fuzzy MNNs. The projective synchronization that is fix-time and with discrete-time delay was investigated in [19]. Considering the lag factor of the system, the projective synchronization was investigated in [20]. Nevertheless, in the synchronization mechanism, parameter mismatch is unavoidable. The projective synchronization of delayed NNs with parameter-mismatched has been studied in [21]. In [22], the quasi-projective synchronization of the parameter-mismatched complex-valued NNs was realized under a feedback controller that is linear. Later on, in [23], the impulsive effect on weak projective synchronization in parameter-mismatched MNNs has been investigated. Uncertainty has not been taken into account in previous studies. We found found that different models ought to be described to meet the practical requirements in the complex situation. Therefore, time-varying delays and uncertainties term in the modeling of MNNs are more essential [24,25,26]. The authors, therefore, have been inspired to go for the model that is less conservative to explore the projective synchronization of MNNs.

To date, in order to realize the information storage and transmission, the multi-links complex NNs were proposed in [27], which are divided into some subnets according to various time-varying delays. They are more realistic than signal link neural networks in the fields of transportation networks [28], social networks [29], brain networks [30], etc. This means that different path transmission delays represent various sub-nets, reflecting the flexibility and universality of the system. In view of these discussions, the muti-links memristive neural networks are more reasonable and have a practical significance in synchronization mechanisms; the authors have adopted coupled MNNs with multi-links. The synchronization with finite time lag of multi-links MNNs under the adaptive control scheme was studied in [31]. Zhao et al. [32] discussed the exponential synchronization of coupled MNNs with multi-links. As far as we know, the dynamic behaviors of multi-links coupled MNNs with time-varying delays and uncertainties remain untouched, especially in dealing with projective synchronization under parameter mismatch situations. Thus, this is the main inspiration for this paper.

For a secure communication mechanism, the plaintexts among the drive and response systems are sent via switching back and forth continuously. The error system stability will greatly affect the quality of signal decryption [33]. Through the event-triggered scheme, quantized synchronization was used in secure communication. During the process of chaotic secure communication, whether the chaos system is stable or not plays a crucial part in the performance of signal processing. As a consequence, ref. [34] recovered the image signals through stability analyses and deep learning methods. From the perspective of synchronization, adaptive synchronization [35] and impulsive synchronization [36] were applied to the security of networks and image encryption, respectively. Neural cryptography was used in image sharing [37] and the synchronization of MNNs was investigated to realize secure communication [38]. Accordingly, the chaotic states of NNs with fractional order can also be employed in signal processing and image encryption [39]. As mentioned above, the projective synchronization mechanism of the proposed system is studied, and its performance regarding signal encryption/decryption of secure communication is investigated in this paper.

Inspired by the above consideration, the aim of this paper is to guarantee projective synchronization of multi-links coupled MNNs with delays that are time-varying and with uncertainties. The following are the novel aspects of current article:Unlike previous coupled MNNs, the proposed model takes the time-varying delays, uncertainties, and multi-links into consideration, which is a class of uncertain switching systems, and it is more helpful to verify the dynamic behavior of systems under different communication situations.The principle of extended comparison and a new approach are proposed to deal with the issue of parameters that are mismatched. To transform the state-depended-coupled MNNs into a class of systems with interval parameters, the criteria of projective synchronization are derived under the mechanism of the novel Lyapunov–Krasovskii functional (LKF). Accordingly, less conservative results compared with the traditional approaches are obtained in this paper. Moreover, the obtained outcomes can be easily extended to various synchronization schemes, depending on the projective parameter.Considering the concept of synchronization, the chaotic sequences of drive and response systems are employed in signal encryption and decryption of secure communication. Taking the advantage of projective synchronization into account, the adaptive signal processing scheme is designed, and the keyspace can expand effectively compared with the conventional methods.

The arrangement of the remaining current study is illustrated below. Section 2 introduces the mathematical model of coupled MNNs, the useful definitions, considerations, and lemmas. Section 3 illustrates the essential outcomes of the current study, involving three theorems and one corollary. Numerical examples are presented to evaluate the correctness of the presented outcomes in Section 4. Finally, the conclusions are given in Section 5.

## 2. Model and Preliminaries

### 2.1. Coupled MNNs Model

Considering with the characteristic of the memristor Figure 1 a class of delayed MNNs are expressed by the following equation
(1)$$\begin{array}{cc} c_{p}\frac{dx_{p}\left(t\right)}{dt} & =-\left[\left(M_{fpg}+N_{hpg}\right)+\frac{1}{R_{p}}\right]x_{p}\left(t\right)+\sum_{g=1}^{n}M_{fpg}\times {\mathrm{sgn}}_{pg}f_{g}\left(x_{g}\left(t\right)\right)\\ \; & +\sum_{g=1}^{n}N_{hpg}\times {\mathrm{sgn}}_{pg}f_{g}\left(x_{g}\left(t-\tau \left(t\right)\right)\right)+I_{p}\left(t\right), \end{array}$$
where the voltage of capacitor is represented as $c_{p}$, and the neural feedback functions with non-delayed and time-varying delays are $f_{g}\left(x_{g}\left(t\right)\right)$ and $f_{g}\left(x_{g}\left(t-\tau \left(t\right)\right)\right)$. Consider the real behavior of solid-state memristors or emulators, $M_{fpg}$ illustrates the memristance of the memristor between $f_{g}\left(x_{g}\left(t\right)\right)$ and $x_{p}\left(t\right)$, then, $N_{hpg}$ represents the memristance of the memristor between $f_{g}\left(x_{g}\left(t-\tau \left(t\right)\right)\right)$ and $x_{g}\left(t-\tau \left(t\right)\right)$. Accordingly, τ(t) denotes the time-varying delay, $R_{p}$ is the resistor, and $I_{p}\left(t\right)$ is an external bias.

For the propose of simplifying the mathematical model of the memristor on the premise of obtaining the pinched hysteresis feature, we select a surrogate memristor model.
(2)$$\begin{array}{cc} \frac{dx_{p}\left(t\right)}{dt} & =-d_{p}\left(x_{p}\left(t\right)\right)x_{p}\left(t\right)+\sum_{g=1}^{n}a_{pg}\left(x_{p}\left(t\right)\right)f_{g}\left(x_{g}\left(t\right)\right)\\ \; & +\sum_{g=1}^{n}b_{pg}\left(x_{p}\left(t-\tau \left(t\right)\right)\right)f_{g}\left(x_{g}\left(t-\tau \left(t\right)\right)\right)+I_{p}\left(t\right), \end{array}$$
where $d_{p}\left(x_{p}\left(t\right)\right)$ is the pth neuron self-inhibition, which represents as $\frac{1}{c_{p}}\left[\left(M_{fpg}+N_{hpg}\right)+\frac{1}{R_{p}}\right]$. $a_{pg}\left(x_{p}\left(t\right)\right)$ and $b_{pg}\left(x_{p}\left(t-\tau \left(t\right)\right)\right)$ represent the memristive connection weights, which denotes as $\frac{1}{c_{p}}\sum_{g=1}^{n}M_{fpg}\times {\mathrm{sgn}}_{pg}$ and $\frac{1}{c_{p}}\sum_{g=1}^{n}N_{hpg}\times {\mathrm{sgn}}_{pg}$; and $f_{g}\left(\cdot \right)$ is the activation function. We assume that the solution $x\left(t\right)={\left(x_{1}\left(t\right),x_{2}\left(t\right),\ldots ,x_{p}\left(t\right)\right)}^{T}$ of system (Equation 2) with initial conditions $x\left(s\right)=\phi \left(s\right)={\left(\phi_{1}\left(t\right),\phi_{2}\left(t\right),\ldots ,\phi_{p}\left(t\right)\right)}^{T}\in C\left(\left[-\tau ,0\right],R^{T}\right)$ exists.

Regarding the characters of a memristor and simplified mathematical model of the memristor Figure 2, the parameters of systems are supposed to meet the following conditions
$$\begin{array}{c} d_{p}\left(x_{p}\left(t\right)\right)=\left\{\begin{array}{cc} {\overset{´}{d}}_{p}, & |x_{p}\left(t\right)|\geq T_{p},\\ \mathrm{unchanged}, & |x_{p}\left(t\right)|=T_{p},\\ {\overset{`}{d}}_{p}, & |x_{p}\left(t\right)|>T_{p}, \end{array}\right. \end{array}$$
$$\begin{array}{c} a_{pg}\left(x_{p}\left(t\right)\right)=\left\{\begin{array}{cc} {\overset{´}{a}}_{pg}, & |x_{p}\left(t\right)|\geq T_{p},\\ \mathrm{unchanged}, & |x_{p}\left(t\right)|=T_{p},\\ {\overset{`}{a}}_{pg}, & |x_{p}\left(t\right)|>T_{p}, \end{array}\right. \end{array}$$
$$\begin{array}{c} b_{pg}\left(x_{p}\left(t\right)\right)=\left\{\begin{array}{cc} {\overset{´}{b}}_{pg}, & |x_{p}\left(t\right)|\geq T_{p},\\ \mathrm{unchanged}, & |x_{p}\left(t\right)|=T_{p},\\ {\overset{`}{b}}_{pg}, & |x_{p}\left(t\right)|>T_{p}. \end{array}\right. \end{array}$$

Consider of *N* time-varying uncertain coupled MNNs as follows
(3)$$\begin{array}{cc} \frac{dx_{i}\left(t\right)}{dt} & =-D\left(t\right)x_{i}\left(t\right)+\left[A\left(t\right)+A_{1}\left(t\right)\right]f\left(x_{i}\left(t\right)\right)\\ \; & +\left[B\left(t\right)+B_{1}\left(t\right)\right]f\left(x_{i}\left(t-\tau_{0}\left(t\right)\right)\right)+\sigma \sum_{j=1}^{N}w_{ij}^{0}\Gamma x_{j}\left(t\right)\\ \; & +\sigma \sum_{j=1}^{N}w_{ij}^{1}\Gamma x_{j}\left(t-\tau_{1}\left(t\right)\right)+I_{i}\left(t\right),i=1,2,...,N, \end{array}$$
then, the delayed coupled MNNs as drive system contains *N* identical MNNs is described as Equation (Equation 3), in which $x_{i}\left(t\right)={\left(x_{i1}\left(t\right),x_{i2}\left(t\right),\ldots ,x_{in}\left(t\right)\right)}^{T}\in R^{n}$ are the state variables of the i−th dynamical networks. $\tau_{0}\left(t\right)$ and $\tau_{1}\left(t\right)$ are time-varying delays, which satisfy ${\dot{\tau }}_{0}\left(t\right)\leq \tau_{0}<1$, ${\dot{\tau }}_{1}\left(t\right)\leq \tau_{1}<1$, $\tau =\max\{\tau_{0},\tau_{1}\}$. We make the uncertainties as $A_{1}\left(t\right)$ and $B_{1}\left(t\right)$. Then, $w_{ij}^{0}$ is non-delayed coupled matrix, and $w_{ij}^{1}$ is delayed coupled matrix. Additionally, we make $w_{ii}^{0}=-\sum_{j=1,j\neq i}^{N}w_{ij}^{0}$ and $w_{ii}^{1}=-\sum_{j=1,j\neq i}^{N}w_{ij}^{1}$, and the following conditions are satisfied [32]
$$\begin{array}{c} \left\{\begin{array}{cc} w_{ij}^{0}=1,w_{ij}^{1}=0 & a\mathrm{directed}\mathrm{edge}\mathrm{from}i\mathrm{to}j,\\ w_{ij}^{0}=0,w_{ij}^{1}=1 & \mathrm{others}. \end{array}\right. \end{array}$$

The constant σ>0 stands for the coupling strength, the matrix $\Gamma =\left(r_{ij}\right)\in R^{n\times n}$ is non-delayed inner connecting matrix, and the coupled matrix $W={\left(w_{ij}\right)}_{N\times N}$ represents the topology structure of the whole networks.

Then, the following state-space equations represent the corresponding response system:(4)$$\begin{array}{cc} \frac{dy_{i}\left(t\right)}{dt} & =-D\left(t\right)y_{i}\left(t\right)+\left[A\left(t\right)+A_{2}\left(t\right)\right]f\left(y_{i}\left(t\right)\right)\\ \; & +\left[B\left(t\right)+B_{2}\left(t\right)\right]f\left(y_{i}\left(t-\tau_{0}\left(t\right)\right)\right)+\sigma \sum_{j=1}^{N}w_{ij}^{0}\Gamma y_{j}\left(t\right)\\ \; & +\sigma \sum_{j=1}^{N}w_{ij}^{1}\Gamma y_{j}\left(t-\tau_{1}\left(t\right)\right)+U_{i}\left(t\right)+I_{i}\left(t\right),i=1,2,...,N, \end{array}$$
where $y_{i}\left(t\right)$ stands for the state vector with initial condition $y\left(s\right)=\varphi \left(s\right)=(\varphi_{1}\left(t\right),\varphi_{2}\left(t\right),\ldots ,$$\varphi_{p}{\left(t\right))}^{T}\in C\left(\left[-\tau ,0\right],R^{T}\right)$. $A_{2}\left(t\right)$ and $B_{2}\left(t\right)$ are the response uncertainties and $U_{i}\left(t\right)$ describes the nonlinear controller that will be constructed later. We choose the parameters similar to the drive system, that is,
(5)$$\begin{array}{cc} \; & \hat{D}={\left({\overset{´}{d}}_{ip}\right)}_{n\times n},\;\overset{ˇ}{D}={\left({\overset{`}{d}}_{ip}\right)}_{n\times n},\;\hat{A}={\left({\overset{´}{a}}_{pg}\right)}_{n\times n},\\ \; & \overset{ˇ}{A}={\left({\overset{`}{a}}_{pg}\right)}_{n\times n},\hat{B}={\left({\overset{´}{b}}_{pg}\right)}_{n\times n},\;\overset{ˇ}{B}={\left({\overset{`}{b}}_{pg}\right)}_{n\times n}. \end{array}$$

Therefore, the drive-system (Equation 3) is modified as
(6)$$\begin{array}{cc} \frac{dx(t)}{dt} & \in -\left(D+co\left[-\tilde{D},\tilde{D}\right]\right)x\left(t\right)+\left[A+co\left[-\tilde{A},\tilde{A}\right]+A_{1}\left(t\right)\right]f\left(x\left(t\right)\right)\\ \; & +\left[B+co\left[-\tilde{B},\tilde{B}\right]+B_{1}\left(t\right)\right]f\left(x\left(t-\tau_{0}\left(t\right)\right)\right)+\sigma W_{0}\Gamma x\left(t\right)\\ \; & +\sigma W_{1}\Gamma x\left(t-\tau_{1}\left(t\right)\right)+I\left(t\right),i=1,2,...,N, \end{array}$$

Then, we obtain
(7)$$\begin{array}{cc} \; & D^{*}=\max\left\{\hat{D},\overset{ˇ}{D}\right\},D^{**}=\min\left\{\hat{D},\overset{ˇ}{D}\right\},A^{*}=\max\left\{\hat{A},\overset{ˇ}{A}\right\},\\ \; & A^{**}=\min\left\{\hat{A},\overset{ˇ}{A}\right\},B^{*}=\max\left\{\hat{B},\overset{ˇ}{B}\right\},B^{**}=\min\left\{\hat{B},\overset{ˇ}{B}\right\},\\ \; & D=\frac{1}{2}\left(D^{*}+D^{**}\right),\tilde{D}=\frac{1}{2}\left(D^{*}-D^{**}\right),A=\frac{1}{2}\left(A^{*}+A^{**}\right),\\ \; & \tilde{A}=\frac{1}{2}\left(A^{*}-A^{**}\right),B=\frac{1}{2}\left(B^{*}+B^{**}\right),\tilde{B}=\frac{1}{2}\left(B^{*}-B^{**}\right). \end{array}$$
where $x\left(t\right)={\left(x_{1}^{T}\left(t\right),x_{2}^{T}\left(t\right),\ldots ,x_{N}^{T}\left(t\right)\right)}^{T}$, $f\left(x\left(t\right)\right)={\left(f_{1}^{T}\left(t\right)\left(x_{1}\left(t\right)\right),f_{1}^{T}\left(t\right)\left(x_{2}\left(t\right)\right),\ldots ,f_{1}^{T}\left(t\right)\left(x_{N}\left(t\right)\right)\right)}^{T}$, $W_{0}={\left(w_{ij}^{0}\right)}_{N\times N}$, $W_{1}={\left(w_{ij}^{1}\right)}_{N\times N}$.

We define the inner couple matrix Γ as identity matrix. The measurable functions are
(8)$$\begin{array}{c} D\left(t\right)=\left\{\begin{array}{cc} \tilde{D}, & D=\bar{D}\\ -\tilde{D}, & D=\underline{D}, \end{array}\right. \end{array}$$
(9)$$\begin{array}{c} A\left(t\right)=\left\{\begin{array}{cc} \tilde{A}, & A=\bar{A}\\ -\tilde{A}, & A=\underline{A} \end{array}\right. \end{array}$$
(10)$$\begin{array}{c} B\left(t\right)=\left\{\begin{array}{cc} \tilde{B}, & B=\bar{B}\\ -\tilde{B}, & B=\underline{B} \end{array}\right. \end{array}$$

Consequently, from the set-valued mappings technique and differential inclusion theory, the Equation (Equation 6) can be transferred as:(11)$$\begin{array}{cc} \frac{dx(t)}{dt} & =-\left(D+D\left(t\right)\right)x\left(t\right)+\left[A+A\left(t\right)+A_{1}\left(t\right)\right]f\left(x\left(t\right)\right)\\ \; & +\left[B+B\left(t\right)+B_{1}\left(t\right)\right]f\left(x\left(t-\tau_{0}\left(t\right)\right)\right)+\sigma W_{0}\Gamma x\left(t\right)\\ \; & +\sigma W_{1}\Gamma x\left(t-\tau_{1}\left(t\right)\right)+I\left(t\right),i=1,2,...,N. \end{array}$$

**Remark** **1.**
*The interval parameters are introduced to transform Equations (Equation 3)–(Equation 11). Accordingly, considering the state dependence characters of the memristor, we derived measurable function Equations (8)–(10) to confirm the variables D(t), A(t) and B(t) cannnot get the maximum and minimum values simultaneously. In other words, when $\hat{D}>\overset{ˇ}{D}$ or $\hat{D}<\overset{ˇ}{D}$, the corresponding $\hat{A}>\overset{ˇ}{A}$ or $\hat{A}<\overset{ˇ}{A}$, $\hat{B}>\overset{ˇ}{B}$ or $\hat{B}<\overset{ˇ}{B}$ may not be satisfied.*


Similarly, we have the response system as
(12)$$\begin{array}{cc} \frac{dy(t)}{dt} & =-\left(D+D^{*}\left(t\right)\right)y\left(t\right)+\left[A+A^{*}\left(t\right)+A_{2}\left(t\right)\right]f\left(y\left(t\right)\right)\\ \; & +\left[B+B^{*}\left(t\right)+B_{2}\left(t\right)\right]f\left(y\left(t-\tau_{0}\left(t\right)\right)\right)+\sigma W_{0}\Gamma y\left(t\right)\\ \; & +\sigma W_{1}\Gamma y\left(t-\tau_{1}\left(t\right)\right)+I\left(t\right),i=1,2,...,N, \end{array}$$
and $D^{*}\left(t\right)\in \left[D^{*},D^{**}\right]$, $A^{*}\left(t\right)\in \left[A^{*},A^{**}\right]$, $B^{*}\left(t\right)\in \left[B^{*},B^{**}\right]$.

### 2.2. Some Useful Definitions and Assumptions

**Assumption** **1.**
*The parameters D(t), A(t), B(t), $A^{*}\left(t\right)$, $B^{*}\left(t\right)$ are norm-bounded and time-varying, which satisfy*

(13)
$$\begin{array}{cc} \; & A\left(t\right)=H_{1}F_{1}\left(t\right)M_{1},A^{*}\left(t\right)=H_{1}E_{1}\left(t\right)M_{1},\\ \; & B\left(t\right)=H_{2}F_{2}\left(t\right)M_{2},B^{*}\left(t\right)=H_{2}E_{2}\left(t\right)M_{2}, \end{array}$$

*we set $M_{1}$ equals to $A^{*}-A^{**}$, and $M_{2}$ equals to $B^{*}-B^{**}$. $H_{1}=H_{2}=\mathrm{diag}\left\{0.5,0.5,\cdots ,0.5\right\}$. For measurable functions $F_{l}\left(t\right)$ and $E_{l}\left(t\right)$ (l=1,2), we have $F_{l}^{T}\left(t\right)F_{l}\left(t\right)\leq I$, $E_{l}^{T}\left(t\right)E_{l}\left(t\right)\leq I$.*


**Assumption** **2.**
*The activation function f(x) is bounded and Lipschitez-continuous. There is a real constant L>0 such that*

(14)
$$\begin{array}{c} |f_{i}\left(\xi_{1}\right)-f_{i}\left(\xi_{2}\right)\left|\leq L\right|\xi_{1}-\xi_{2}|, \end{array}$$

*for all $\xi_{1}\in R$, $\xi_{2}\in R$, i∈N, f(0)=0.*


**Assumption** **3.**
*For all initial values $x\left(s\right)=\phi \left(s\right)={\left(\phi_{1}\left(t\right),\phi_{2}\left(t\right),\ldots ,\phi_{p}\left(t\right)\right)}^{T}\in C\left(\left[-\tau ,0\right],R^{T}\right)$, the solution of (Equation 11) is bounded, that is $|x_{i}\left(t\right)|\leq M$ for t∈[−τ,∞).*


**Assumption** **4.**
*Supposing universities are bounded, which means $|A_{l}\left(t\right)|\leq Z$, $|B_{l}\left(t\right)|\leq Z$ (l=1,2), and satisfy*

(15)
$$\begin{array}{cc} \; & A_{1}\left(t\right)=A_{1}\omega_{A}\left(t\right),B_{1}\left(t\right)=B_{1}\omega_{B}\left(t\right),\\ \; & A_{2}\left(t\right)=A_{2}\mu_{A}\left(t\right),B_{2}\left(t\right)=B_{2}\mu_{B}\left(t\right), \end{array}$$

*and $\omega_{A}^{2}\left(t\right)\leq 1$, $\omega_{B}^{2}\left(t\right)\leq 1$, $\mu_{A}^{2}\left(t\right)\leq 1$, and $\mu_{B}^{2}\left(t\right)\leq 1$.*


**Lemma** **1.**
*If P and Q represent the matrices with suitable dimensions, such that, such as*

(16)
$$\begin{array}{c} P^{T}Q+Q^{T}P\leq \delta P^{T}P+\frac{1}{\delta }Q^{T}Q, \end{array}$$

*which δ>0.*


**Definition** **1.**
*If there exist constant α>0, that is, for any colution x(t) and y(t) of (Equation 11) and (Equation 12), one can derive*

(17)
$$\begin{array}{c} \lim_{t\to +\infty }||y\left(t\right)-\alpha x\left(t\right)||=0, \end{array}$$

*then, systems (Equation 11) and (Equation 12) can achieve projective synchronization.*


Notably, the error system $e\left(t\right)={\left(e_{1}^{T}\left(t\right),e_{2}^{T}\left(t\right),\ldots ,e_{N}^{T}\left(t\right)\right)}^{T}$ such that e(t)=y(t)−αx(t) can be extracted from relations (Equation 11) and (Equation 12) as follows
(18)$$\begin{array}{cc} \frac{de(t)}{dt} & =-\left(D+D^{*}\left(t\right)\right)e\left(t\right)+\left[A+A^{*}\left(t\right)+A_{2}\left(t\right)\right]f\left(e\left(t\right)\right)\\ \; & +\left[B+B^{*}\left(t\right)+B_{2}\left(t\right)\right]f\left(e\left(t-\tau_{0}\left(t\right)\right)\right)+\Upsilon \left(t\right)\\ \; & +\sigma W_{0}\Gamma e\left(t\right)+\sigma W_{1}\Gamma e\left(t-\tau_{1}\left(t\right)\right)+U\left(t\right), \end{array}$$
where $\Upsilon \left(t\right)=\left[A+A^{*}\left(t\right)+A_{2}\left(t\right)\right]f\left(\alpha x\left(t\right)\right)+\left[B+B^{*}\left(t\right)+B_{2}\left(t\right)\right]f\left(\alpha x\left(t-\tau_{0}\left(t\right)\right)\right)+$$\alpha \left(D^{*}\left(t\right)-D\left(t\right)\right)x\left(t\right)-\alpha \left[A+A\left(t\right)+A_{1}\left(t\right)\right]f\left(x\left(t\right)\right)-\alpha \left[B+B\left(t\right)+B_{1}\left(t\right)\right]f\left(x\left(t-\tau_{0}\left(t\right)\right)\right)$, f(e(t))=f(y(t))−f(αx(t)), $f\left(e\left(t-\tau_{0}\left(t\right)\right)\right)=f\left(y\left(t\right)\right)-f\left(\alpha x\left(t-\tau_{0}\left(t\right)\right)\right)$.

**Remark** **2.**
*If α>0, systems (Equation 11) and (Equation 12) can achieve synchronization; if α<0, systems (Equation 11) and (Equation 12) can achieve anti-synchronization; if α=0, system (Equation 11) is stable to the origin state asymptotically.*


## 3. Fudamental Results

The current section presents the solution to the adaptive synchronization problem of uncertain coupled MNNs under the nonlinear control and its application.

Before starting our fundamental outcomes, the controller is designed as follows
(19)U(t)=−Re(t)−Ωsign(e(t))
where $R=\mathrm{diag}\left(r_{i1},r_{i2},\ldots ,r_{in}\right)\left(i=1,2,\ldots ,N\right)$ stands for the feedback controller matrix that will be constructed. $\Omega =\mathrm{diag}(w_{i1},w_{i2},\ldots ,w_{in})$, $r_{ip}$ and $w_{ip}$, p=1,2,…,n describe positive constants, and
(20)$$\begin{array}{cc} \omega_{ip} & =\sum_{l=0}^{n}2L\alpha (|{\overset{´}{d}}_{ip}-{\overset{`}{d}}_{ip}\left|+\right|{\overset{´}{a}}_{pg}-{\overset{`}{a}}_{pg}\left|+\right|{\overset{´}{b}}_{pg}-{\overset{`}{b}}_{pg}\left|+2Z\right)M. \end{array}$$

**Theorem** **1.**
*Consider Assumptions 1–4; systems (Equation 11) and (Equation 12) are projective synchronization with the control law (Equation 19), if there exist constants $\theta_{1}>0$, $\theta_{2}>0$, such that*



(21)
$$\begin{array}{cc} \Lambda & =2D-2\tilde{D}+\theta_{1}AA^{T}+\frac{3L^{2}}{\theta_{1}}+\theta_{1}\left|\right|M_{1}{\left|\right|}^{2}H_{1}^{T}H_{1}+\theta_{1}\left|\right|A_{2}{\left|\right|}^{2}H_{1}^{T}A_{2}\\ \; & +\theta_{2}BB^{T}+\theta_{2}\left|\right|M_{2}{\left|\right|}^{2}H_{2}^{T}H_{2}+\theta_{2}\left|\right|B_{2}{\left|\right|}^{2}H_{1}^{T}B_{2}+\sigma^{2}\Gamma W_{1}W_{1}^{T}\Gamma^{T}\\ \; & +\sigma W_{0}\Gamma -2R<0. \end{array}$$


**Proof.** Assume the subsequent nonnegative function for system (Equation 18)
(22)$$\begin{array}{cc} V(t) & =e^{T}\left(t\right)e\left(t\right)+\frac{1}{1-\tau }[\frac{3}{\theta_{2}}L^{2}\int_{t-\tau_{0}\left(t\right)}^{t}e^{T}\left(s\right)e\left(s\right)ds\\ \; & +\int_{t-\tau_{1}\left(t\right)}^{t}e^{T}\left(s\right)e\left(s\right)ds]. \end{array}$$
We make
(23)$$\begin{array}{cc} V_{1}\left(t\right) & =e^{T}\left(t\right)e\left(t\right),\\ V_{2}\left(t\right) & =\frac{1}{1-\tau }\left[\frac{3}{\theta_{2}}L^{2}\int_{t-\tau_{0}\left(t\right)}^{t}e^{T}\left(s\right)e\left(s\right)ds+\int_{t-\tau_{1}\left(t\right)}^{t}e^{T}\left(s\right)e\left(s\right)ds\right]. \end{array}$$Based on the trajectory of e(t), we get the derivative of $V_{1}\left(t\right)$ as follows
(24)$$\begin{array}{cc} {\dot{V}}_{1}\left(t\right) & =2e^{T}\left(t\right)\dot{e}\left(t\right)\\ \; & =2e^{T}\left(t\right)[-\left(D+D^{*}\left(t\right)\right)e\left(t\right)+\Upsilon \left(t\right)+Af\left(e\left(t\right)\right)+A^{*}\left(t\right)f\left(e\left(t\right)\right)\\ \; & +A_{2}\left(t\right)f\left(e\left(t\right)\right)+Bf\left(e\left(t-\tau_{0}\left(t\right)\right)\right)+B^{*}\left(t\right)f\left(e\left(t-\tau_{0}\left(t\right)\right)\right)\\ \; & +B_{2}\left(t\right)f\left(e\left(t-\tau_{0}\left(t\right)\right)\right)+\sigma W_{0}\Gamma e\left(t\right)+\sigma W_{1}\Gamma e\left(t-\tau_{1}\left(t\right)\right)\\ \; & -Re(t)-\Omega \mathrm{sign}(e(t))]. \end{array}$$According to Assumptions 1–3 and Lemma 1, we deduce
(25)$$\begin{array}{c} 2e^{T}\left(t\right)\left(D+D^{*}\left(t\right)\right)e\left(t\right)\leq 2\left(D-\tilde{D}\right)e^{T}\left(t\right)e\left(t\right). \end{array}$$
(26)$$\begin{array}{cc} \; & 2e^{T}\left(t\right)\left[A+A^{*}\left(t\right)+A_{2}\left(t\right)\right]f\left(e\left(t\right)\right)\\ \; & \leq \theta_{1}e^{T}\left(t\right)AA^{T}e\left(t\right)+\frac{1}{\theta_{1}}f^{T}\left(e\left(t\right)\right)f\left(e\left(t\right)\right)+\theta_{1}e^{T}\left(t\right)A^{*}\left(t\right){\left(A^{*}\left(t\right)\right)}^{T}e\left(t\right)\\ \; & +\frac{1}{\theta_{1}}f^{T}\left(e\left(t\right)\right)f\left(e\left(t\right)\right)+\theta_{1}e^{T}\left(t\right)A_{2}\left(t\right)\Delta A_{2}^{T}\left(t\right)e\left(t\right)+\frac{1}{\theta_{1}}f^{T}\left(e\left(t\right)\right)f\left(e\left(t\right)\right)\\ \; & \leq \theta_{1}e^{T}\left(t\right)AA^{T}e\left(t\right)+\frac{3}{\theta_{1}}f^{T}\left(e\left(t\right)\right)f\left(e\left(t\right)\right)\\ \; & +\theta_{1}e^{T}\left(t\right)M_{1}^{T}E_{1}^{T}\left(t\right)H_{1}^{T}H_{1}E_{1}\left(t\right)M_{1}\left(t\right)e\left(t\right)+\theta_{1}e^{T}\left(t\right)\mu_{A}^{T}\left(t\right)A_{2}^{T}A_{2}\mu_{A}\left(t\right)e\left(t\right)\\ \; & \leq e^{T}\left(t\right)\left(\theta_{1}e^{T}\left(t\right)AA^{T}+\frac{3}{\theta_{1}}L^{2}+\theta_{1}\left|\right|M_{1}{\left|\right|}^{2}H_{1}^{T}H_{1}+\theta_{1}A_{2}^{T}A_{2}\right)e\left(t\right). \end{array}$$Similarly, we get
(27)$$\begin{array}{cc} \; & 2e^{T}\left(t\right)Bf\left(e\left(t-\tau_{0}\left(t\right)\right)\right)+2e^{T}\left(t\right)B^{*}\left(t\right)f\left(e\left(t-\tau_{0}\left(t\right)\right)\right)\\ \; & +2e^{T}\left(t\right)B_{2}\left(t\right)f\left(e\left(t-\tau_{0}\left(t\right)\right)\right)\\ \; & \leq e^{T}\left(t\right)\theta_{2}e^{T}\left(t\right)BB^{T}e\left(t\right)+e^{T}\left(t\right)\theta_{2}\left|\right|M_{2}{\left|\right|}^{2}H_{2}^{T}H_{2}e\left(t\right)\\ \; & +e^{T}\left(t\right)\theta_{2}B_{2}^{T}B_{2}e\left(t\right)+\frac{3}{\theta_{2}}L^{2}e^{T}\left(t-\tau_{0}\left(t\right)\right)e\left(t-\tau_{0}\left(t\right)\right). \end{array}$$
By usint Young’s inequality and Lemma 1, we have
(28)$$\begin{array}{c} 2e^{T}\left(t\right)\sigma W_{1}\Gamma e\left(t-\tau_{1}\left(t\right)\right)\\ \leq \sigma^{2}e^{T}\left(t\right)\Gamma W_{1}W_{1}^{T}e\left(t\right)+e^{T}\left(t-\tau_{1}\left(t\right)\right)e\left(t-\tau_{1}\left(t\right)\right). \end{array}$$For Υ(t), we have
(29)$$\begin{array}{cc} \; & |\left[A+A^{*}\left(t\right)+A_{2}\left(t\right)\right]f\left(\alpha x\left(t\right)\right)|\\ \; & \leq |A+\tilde{A}+Z||f\left(\alpha x\left(t\right)\right)|\leq \alpha L\left(A+\tilde{A}+Z\right)M. \end{array}$$Then we get
(30)$$\begin{array}{cc} \; & |\left[B+B^{*}\left(t\right)+B_{2}\left(t\right)\right]f\left(\alpha x\left(t-\tau \left(0\right)\left(t\right)\right)\right)|\leq \alpha L\left(B+\tilde{B}+Z\right)M,\\ \; & \alpha |\left(D^{*}\left(t\right)-D\left(t\right)\right)x\left(t\right)|\leq 2\alpha \tilde{D}M, \end{array}$$
and
(31)$$\begin{array}{cc} \; & \alpha |\left[A+A\left(t\right)+A_{1}\left(t\right)\right]f\left(x\left(t\right)\right)|\geq \alpha L\left(A-\tilde{A}-Z\right)M,\\ \; & \alpha |\left[B+B\left(t\right)+B_{1}\left(t\right)\right]f\left(\alpha x\left(t-\tau \left(0\right)\left(t\right)\right)\right)|\geq \alpha L\left(B-\tilde{B}-Z\right)M. \end{array}$$Consider (Equation 18) and mentioned above, we conclude
(32)$$\begin{array}{cc} \; & \Upsilon \left(t\right)\leq 2\alpha L\left(\tilde{A}+\tilde{B}\tilde{D}+2Z\right)M. \end{array}$$Then, we take U(t) into account for deduce
(33)$$\begin{array}{c} 2e^{T}\left(t\right)U\left(t\right)=-2Re^{T}\left(t\right)e\left(t\right)-2e^{T}\left(t\right)\Omega \mathrm{sign}\left(e\left(t\right)\right), \end{array}$$
then we make
(34)$$\begin{array}{cc} \; & 2e^{T}\left(t\right)\left(\Upsilon \left(t\right)-\Omega \mathrm{sign}\left(e\left(t\right)\right)\right)\\ \; & \leq 2\sum_{p=0}^{n}\left|e_{ip}\left(t\right)\right|\left[2\alpha L\sum_{g=0}^{n}\left({\tilde{a}}_{pg}+{\tilde{b}}_{pg}+{\tilde{d}}_{pg}+2Z\right)M-\omega_{ip}\right]=0. \end{array}$$Similarly, the derivative of $V_{2}\left(t\right)$ is described as
(35)$$\begin{array}{cc} {\dot{V}}_{2}\left(t\right) & =\frac{3L^{2}}{\left(1-\tau \right)\theta_{2}}e^{T}\left(t\right)e\left(t\right)-e^{T}\left(t-\tau_{1}\left(t\right)\right)e\left(t-\tau_{1}\left(t\right)\right)\\ \; & -\frac{3L^{2}}{\theta_{2}}e^{T}\left(t-\tau_{0}\left(t\right)\right)e\left(t-\tau_{0}\left(t\right)\right) \end{array}$$Combing the mentioned above, we have
(36)$$\begin{array}{c} \dot{V}\left(t\right)\leq e^{T}\left(t\right)\Lambda e\left(t\right). \end{array}$$Accordin to Definition 1, we get $\dot{V}\left(t\right)\leq e^{T}\left(t\right)\Lambda e\left(t\right)<0$, that is, we make Λ<0 can realize the projective synchronization among systems (12) and (13).The proof is now completed. □

**Corollary** **1.**
*As a particular case, we consider that there are no uncertainties between systems (Equation 11) and (Equation 12). Now, the error system is rewritten as follows.*

(37)
$$\begin{array}{cc} \frac{de(t)}{dt} & =-\left(D+D^{*}\left(t\right)\right)e\left(t\right)+U\left(t\right)+\Upsilon \left(t\right)+\left[B+B^{*}\left(t\right)\right]f\left(e\left(t-\tau_{0}\left(t\right)\right)\right)\\ \; & +\left[A+A^{*}\left(t\right)\right]f\left(e\left(t\right)\right)+\sigma W_{0}\Gamma e\left(t\right)+\sigma W_{1}\Gamma e\left(t-\tau_{1}\left(t\right)\right), \end{array}$$

*where $\Upsilon \left(t\right)=\left[A+A^{*}\left(t\right)\right]f\left(\alpha x\left(t\right)\right)+\left[B+B^{*}\left(t\right)\right]f\left(\alpha x\left(t-\tau_{0}\left(t\right)\right)\right)+\alpha \left(D^{*}\left(t\right)-D\left(t\right)\right)x\left(t\right)-\alpha \left[A+A\left(t\right)\right]f\left(x\left(t\right)\right)-\alpha \left[B+B\left(t\right)\right]f\left(x\left(t-\tau_{0}\left(t\right)\right)\right)$, f(e(t))=f(y(t))−f(αx(t)), $f\left(e\left(t-\tau_{0}\left(t\right)\right)\right)=f\left(y\left(t\right)\right)-f\left(\alpha x\left(t-\tau_{0}\left(t\right)\right)\right)$.*


Suppose Assumptions 1–4 hold, the systems systems (Equation 11) and (Equation 12) can achieve projective synchronization with contrl inputs, if the parameters are choosen as $\theta_{1}>0$, $\theta_{2}>0$, which satisfy
(38)$$\begin{array}{cc} \Lambda & =2D-2\tilde{D}+\theta_{1}AA^{T}+\frac{2L^{2}}{\theta_{1}}+\theta_{1}\left|\right|M_{1}{\left|\right|}^{2}H_{1}^{T}H_{1}\\ \; & +\theta_{2}BB^{T}+\theta_{2}\left|\right|M_{2}{\left|\right|}^{2}H_{2}^{T}H_{2}+\sigma^{2}\Gamma W_{1}W_{1}^{T}\Gamma^{T}+\sigma W_{0}\Gamma -2R<0. \end{array}$$

**Proof.** Consider the nonnegative function, for system (Equation 37), the Lyapunov function is proposed as follows
(39)$$\begin{array}{cc} V(t) & =e^{T}\left(t\right)e\left(t\right)+\frac{1}{1-\tau }[\frac{2}{\theta_{2}}L^{2}\int_{t-\tau_{0}\left(t\right)}^{t}e^{T}\left(s\right)e\left(s\right)ds\\ \; & +\int_{t-\tau_{1}\left(t\right)}^{t}e^{T}\left(s\right)e\left(s\right)ds]. \end{array}$$This proof can be obtained immediately by considering the uncertainties as zero in Theorem 1. Hence, it is neglected here. □

**Remark** **3.**
*During the process of synchronization, the control with discontinuous characteristics is essential to the converge of the error system. However, the stochastic disturbance from various uncertainties inevitably disturbs the practical communication among subsystems of coupled neural networks. Therefore, a simple adaptive method is proposed for the projective synchronization of the system (11) based on the designed Lyapunov function.*


**Theorem** **2.**
*Consider Assumptions 1–4, systems (Equation 11) and (Equation 12) are projective synchronization with the adaptive control approach (Equation 41), if there are constants $\theta_{1}>0$, $\theta_{2}>0$, such that*

(40)
$$\begin{array}{cc} \Lambda & =2\left(\frac{1}{1-\tau }+L_{A}-D\right)+2\sigma W_{0}+\sigma^{2}\Gamma W_{1}W_{1}^{T}\Gamma^{T}+\frac{3L^{2}}{\theta_{1}}+\theta_{1}[AA^{T}\\ \; & +\left|\right|M_{1}{\left|\right|}^{2}+\left|\right|M_{1}{\left|\right|}^{2}H_{1}^{T}H_{1}+A^{*}{\left(A^{*}\right)}^{T}]+\theta_{2}[BB^{T}+\left|\right|M_{2}{\left|\right|}^{2}\\ \; & +\left|\right|M_{2}{\left|\right|}^{2}H_{2}^{T}H_{2}+B^{*}{\left(B^{*}\right)}^{T}]<0, \end{array}$$

*and the adaptive controller (Equation 41) as follows.*

(41)
$$\left\{\begin{array}{c} U(t)=-R(t)e(t)-\Omega \mathrm{sign}(e(t))\\ \dot{R}\left(t\right)=R^{*}e^{T}\left(t\right)e\left(t\right) \end{array}\right.$$

*where $R^{*}$ is positive constant.*


**Proof.** The subsequent Lyapunov function is established for the error system
(42)$$\begin{array}{cc} V(t) & =e^{T}\left(t\right)e\left(t\right)+\frac{{\left(R\left(t\right)+L_{A}\right)}^{2}}{R^{*}}\\ \; & \frac{1}{1-\tau }\left[\frac{3}{\theta_{2}}L^{2}\int_{t-\tau_{0}\left(t\right)}^{t}e^{T}\left(s\right)e\left(s\right)ds+\int_{t-\tau_{1}\left(t\right)}^{t}e^{T}\left(s\right)e\left(s\right)ds\right]. \end{array}$$For the first of (Equation 42), we have
(43)$$\begin{array}{cc} {\dot{V}}_{1}\left(t\right) & =2e^{T}\left(t\right)\dot{e}\left(t\right)\\ \; & =2e^{T}\left(t\right)[-\left(D+D^{*}\left(t\right)\right)e\left(t\right)+\Upsilon \left(t\right)+Af\left(e\left(t\right)\right)+A^{*}\left(t\right)f\left(e\left(t\right)\right)\\ \; & +A_{2}\left(t\right)f\left(e\left(t\right)\right)+Bf\left(e\left(t-\tau_{0}\left(t\right)\right)\right)+B^{*}\left(t\right)f\left(e\left(t-\tau_{0}\left(t\right)\right)\right)\\ \; & +B_{2}\left(t\right)f\left(e\left(t-\tau_{0}\left(t\right)\right)\right)+\sigma W_{0}\Gamma e\left(t\right)+\sigma W_{1}\Gamma e\left(t-\tau_{1}\left(t\right)\right)\\ \; & -R\left(t\right)e\left(t\right)-\Lambda \mathrm{sign}\left(e\left(t\right)\right)]+2\left(R\left(t\right)+L_{A}\right)e^{T}\left(t\right)e\left(t\right). \end{array}$$Similar to the proof of Theorem 1, combining with Definition 1, we can obtain
(44)$$\begin{array}{c} \dot{V}\left(t\right)\leq e^{T}\left(t\right)\Lambda e\left(t\right), \end{array}$$
and
(45)$$\begin{array}{cc} \Lambda & =2\left(\frac{1}{1-\tau }+L_{A}-D\right)+2\sigma W_{0}+\sigma^{2}\Gamma W_{1}W_{1}^{T}\Gamma^{T}+\frac{3L^{2}}{\theta_{1}}+\theta_{1}[AA^{T}\\ \; & +\left|\right|M_{1}{\left|\right|}^{2}+\left|\right|M_{1}{\left|\right|}^{2}H_{1}^{T}H_{1}+A^{*}{\left(A^{*}\right)}^{T}]+\theta_{2}[BB^{T}+\left|\right|M_{2}{\left|\right|}^{2}\\ \; & +\left|\right|M_{2}{\left|\right|}^{2}H_{2}^{T}H_{2}+B^{*}{\left(B^{*}\right)}^{T}]<0. \end{array}$$Accordingly, we conclude the gain matrix with adaptive controller (Equation 41) as follows
(46)$$\begin{array}{cc} \omega_{ip} & =\sum_{g=0}^{n}2L\alpha (|{\overset{´}{d}}_{ip}-{\overset{`}{d}}_{ip}\left|+\right|{\overset{´}{a}}_{pg}-{\overset{`}{a}}_{pg}\left|+\right|{\overset{´}{b}}_{pg}-{\overset{`}{b}}_{pg}\left|+2Z\right)M. \end{array}$$That is, with the synchronization criteria (Equation 21) and (Equation 46), systems (Equation 11) and (Equation 12) can realize the complete projective synchronization under the adaptive control approach (Equation 41).The proof is now completed. □

**Theorem** **3.**
*Consider the encrypted chaotic sequences x(t) from drive system (Equation 11) are added to the plaintexts r(t), then we get the transmitted signals s(t) as follows*

(47)
$$\begin{array}{cc} s(t) & =x(t)+r(t). \end{array}$$


*Accordingly, the recovered signal is obtained based on decrypted chaotic y(t) from response system (Equation 12) as the following*

(48)
$$\begin{array}{c} r^{*}\left(t\right)=s\left(t\right)-y\left(t\right). \end{array}$$


*Then, an adaptive observer synchronization of coupled MNNs approach is designed as:*

(49)
$$\begin{array}{cc} \frac{dx(t)}{dt} & =-\left(D+D\left(t\right)\right)x\left(t\right)+\left[A+A\left(t\right)+A_{1}\left(t\right)\right]f\left(x\left(t\right)\right)\\ \; & +\left[B+B\left(t\right)+B_{1}\left(t\right)\right]f\left(x\left(t-\tau_{0}\left(t\right)\right)\right)\\ \; & +\sigma W_{0}\Gamma x\left(t\right)+\sigma W_{1}\Gamma x\left(t-\tau_{1}\left(t\right)\right)\\ \; & +I\left(t\right)+k_{1}\left(r\left(t\right)-w\left(t\right)\right),i=1,2,...,N, \end{array}$$

*and*

(50)
$$\begin{array}{cc} \frac{dy(t)}{dt} & =-\left(D+D^{*}\left(t\right)\right)y\left(t\right)+\left[A+A^{*}\left(t\right)+A_{2}\left(t\right)\right]f\left(y\left(t\right)\right)\\ \; & +\left[B+B^{*}\left(t\right)+B_{2}\left(t\right)\right]f\left(y\left(t-\tau_{0}\left(t\right)\right)\right)\\ \; & +\sigma W_{0}\Gamma y\left(t\right)+\sigma W_{1}\Gamma y\left(t-\tau_{1}\left(t\right)\right)\\ \; & +I\left(t\right)+U^{*}\left(t\right),i=1,2,...,N, \end{array}$$


*where $\dot{w}\left(t\right)=k_{2}\left(r\left(t\right)-w\left(t\right)\right)$.*

*The proposed adaptive controller $U^{*}\left(t\right)$ as follows*

(51)
U∗(t)=U(t)+k1s(t)−y(t)−w∗(t)U(t)=−Re(t)−Ωsign(e(t))w˙∗(t)=k2s(t)−y(t)−w∗(t)

*where $k_{1},k_{2}$ are positive constants.*


## 4. Numerical Simulations

The current section presents various numerical examples to evaluate the correctness of the developed theoretical results.

**Example** **1.**
*Projective synchronization.*


Assume the three-dimensional coupled MNNs (Equation 11) and (Equation 12) consists three neural networks and the edge weights are 1. The topology structure of the system is illustrated as the coupling strength σ=1, the inner connecting matrix Γ, coupled matrix $W_{0}$, and $W_{1}$ are as follows
$$\Gamma =\left(\begin{array}{ccc} 1 & 0 & 0\\ 0 & 1 & 0\\ 0 & 0 & 1 \end{array}\right),W_{0}=W_{1}=\left(\begin{array}{ccc} -2 & 1 & 1\\ 1 & -1 & 0\\ 1 & 0 & -1 \end{array}\right).$$

For systems (Equation 11) and (Equation 12), the time-varying delay are $\tau_{0}\left(t\right)=e^{t}/\left(e^{t}+1\right)$, and $\tau_{1}\left(t\right)=\tanh\left(t\right)$. The initial values for drive system are $\phi ={\left[-1.57,0.99,1.31,0.37,-1.21,0.36,1.11,-1.28,-0.97\right]}^{T}\in C(\left[-\tau ,0\right]$, then we make the intial valued for response system are $\varphi ={\left[0.78,2.11,-0.58,2.04,-0.89,-1.44,-1.67,1.89,-0.78\right]}^{T}\in C(\left[-\tau ,0\right]$. The active function is $f\left(x\right)=\frac{\left|x+1|-|x-1\right|}{2}-1$.

Based on the mismatched parameters for coupled MNNs transmission method, we make the following parameters
$$D=\left(\begin{array}{ccc} 1.25 & 0.9 & 0.9\\ 1.05 & 1.1 & 0.95\\ 1.0 & 1.0 & 1.05 \end{array}\right),\tilde{D}=\left(\begin{array}{ccc} 0.25 & -0.1 & -0.1\\ 0.25 & 0.1 & -0.05\\ 0.1 & 0 & 0.15 \end{array}\right).$$
$$A=\left(\begin{array}{ccc} -1.65 & -2.6 & 2.75\\ 1.1 & 1.6 & -1.4\\ 1.0 & 1.65 & -1.2 \end{array}\right),\tilde{A}=\left(\begin{array}{ccc} 0.15 & -0.2 & -0.25\\ -0.2 & -0.1 & -0.2\\ -0.2 & -0.35 & -0.2 \end{array}\right).$$
$$B=\left(\begin{array}{ccc} 0.35 & 0.25 & 0.8\\ -3.2 & -3.55 & -1.7\\ -4.2 & -2.9 & 0.75 \end{array}\right),\tilde{B}=\left(\begin{array}{ccc} 0.05 & -0.05 & 0.25\\ -0.2 & 0.25 & 0.2\\ 0.1 & 0.1 & -0.25 \end{array}\right).$$

Taken into the following uncertainties into consideration, we have
$$A_{1}\left(t\right)=0.3\cos\left(t\right)\left(\begin{array}{ccc} 1 & 1 & 1\\ 1 & 1 & 1\\ 1 & 1 & 1 \end{array}\right),B_{1}\left(t\right)=-0.1\sin\left(t-\frac{e^{t}}{e^{t}+1}\right)\left(\begin{array}{ccc} 1 & 1 & 1\\ 1 & 1 & 1\\ 1 & 1 & 1 \end{array}\right).$$
$$A_{2}\left(t\right)=-0.2\sin\left(t\right)\left(\begin{array}{ccc} 1 & 1 & 1\\ 1 & 1 & 1\\ 1 & 1 & 1 \end{array}\right),B_{2}\left(t\right)=0.5\cos\left(t-\frac{e^{t}}{e^{t}+1}\right)\left(\begin{array}{ccc} 1 & 1 & 1\\ 1 & 1 & 1\\ 1 & 1 & 1 \end{array}\right).$$

According to Assumptions 1 and 2, we make
$$M_{1}=\left(\begin{array}{ccc} 0.3 & 0.4 & 0.3\\ 0.4 & 0.2 & 0.4\\ 0.4 & 0.7 & 0.4 \end{array}\right),M_{2}=\left(\begin{array}{ccc} 0.2 & 0.2 & 0.3\\ 0.2 & 0.2 & 0.5\\ 0.5 & 0.5 & 0.4 \end{array}\right).$$
$$H_{1}=H_{2}=\left(\begin{array}{ccc} 0.5 & 0 & 0\\ 0 & 0.5 & 0\\ 0 & 0 & 0.5 \end{array}\right),L=\left(\begin{array}{ccc} 1 & 0 & 0\\ 0 & 1 & 0\\ 0 & 0 & 1 \end{array}\right).$$

For drive/response systems (Equation 11) and (Equation 12), we choose three different initial values to illustrate the chaotic for such system, the dynamic trajectories of states without controller are presented in Figure 3. Additionally, Figure 3c shows the input of the ineffective control, which means the states of such system is chaotic under the ineffective control method, and the system will be controlled under the suitabl control methods as following.

For Theorem 1, when projective factor α=1, we have the complete stable of error system. Then we set R=10 as control gains for feedback controller (Equation 19), as illustrated in Figure 4, such controller can guarantee the stability of the system. When projective factor α=−1, we obtain the antisynchronization between systems (Equation 11) and (Equation 12), which shown in Figure 4e.

For Theorem 2, we verify the effect of adaptive controller for such systems. It can be seen from Figure 4f, the rationality and effectiveness of the designed controller (Equation 41)

**Remark** **4.**
*For controller, we define as follows*

(52)
$$\mathrm{sign}\left(e\left(t\right)\right)\left\{\begin{array}{cc} -1, & \;\mathrm{if}\;e(t)<0,\\ 0, & \;\mathrm{if}\;e(t)=0,\\ 1, & \;\mathrm{if}\;e(t)>0, \end{array}\right.$$

*but the characteristic of sign(e(t)) will lead to a buffeting phenomenon; thus we replace $\frac{e(t)}{\;e(t)|+K}\left(K=0.001\right)$ to to alleviate this phenomenon. The Figure 5 illustrates the effective of such method. Figure 6a–c demonstrate the different types of controllers under buffeting phenomenon due to the sign(e(t)) factor, and Figure 6d–f show the corresponding controllers, which get away from the buffeting under the influence of $\frac{e(t)}{\;e(t)|+K}$.*


For Corollary 1, we make the uncertainties as zeros, the states of (Equation 37) without control are presented in Figure 5a–c. It is shown that the error system (Equation 18) converges to zero gradually, which verify the Corollary 1 is reasonable.

**Example** **2.**
*Secure communication.*


The secure communication process of the proposed algorithm is illustrated in Figure 7. We make the same parameters as Corollary 1. Two plaintexts are $r_{1}\left(t\right)=\sin\left(t\right)$ and $r_{2}\left(t\right)=\cos\left(t\right)$. For the process of encryption, two encryption chaotic sequences $x_{1}\left(t\right)$ and $x_{2}\left(t\right)$ are chosen from the drive system (Equation 11). We combine plaintext signals and chaotic sequences, and the transmitted signals as shown in Figure 8. Consider the process of the decryption, two decryption chaotic sequences $y_{1}\left(t\right)$ and $y_{2}\left(t\right)$ are chosen from the corresponding response system (Equation 12). For the adaptive controller (Equation 51), we set $k_{1}=0.03$, $k_{2}=15$. Figure 9 illustrated the trajectories of plaintexts and decrypted signals. From Figure 10, we notice the error between plaintexts and decrypted signals converge to zero, which means the projective synchronization approach can solve the encryption/decryption problem effective in secure communication.

**Remark** **5.**
*Comparing with current literature [40,41] for secure communication, the parameters in the proposed model, that is, $W_{0}$ and $W_{1}$ can not only demonstrate the special structure of MNNs, but also can build the internal connection between the sufficient condition and the projective synchronization approach under the multi-links coupled condition. Consequently, in terms of the effect of secure communication (e.g., number of keys, types of encryption and decryption), the proposed method is more flexible and expandable than other approaches.*


## 5. Conclusions

This paper focused on the projective synchronization of coupled multi-links MNNs with uncertainties and delays that is time-varying. Aiming at the issue of parameter mismatch, the principle of extended comparison and a new approach were employed to transform the proposed system into one with interval parameters. Furthermore, according to the designed LKF, several sufficient criteria for projective synchronization were derived under the nonlinear controller. Finally, the chaotic sequences of drive and response systems were applied in signal encryption and decryption of secure communication. Based on the concept of projective synchronization, the adaptive control mechanism has been proposed to improve the effectiveness of signal decryption and decryption in secure communication.

## Figures and Tables

**Figure 1 entropy-25-01241-f001:**
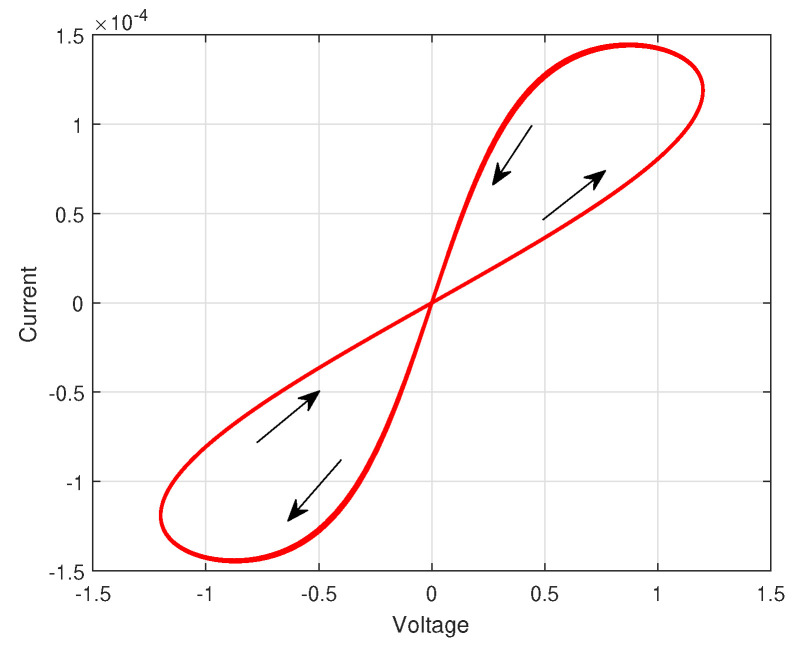
The nonlinear behavioral model of the memristor.

**Figure 2 entropy-25-01241-f002:**
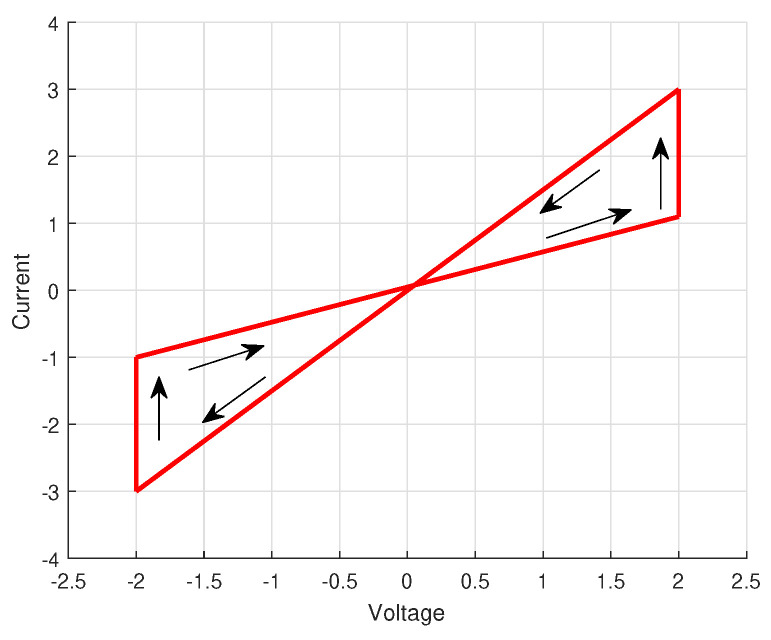
The simplified mathematical model of the memristor.

**Figure 3 entropy-25-01241-f003:**
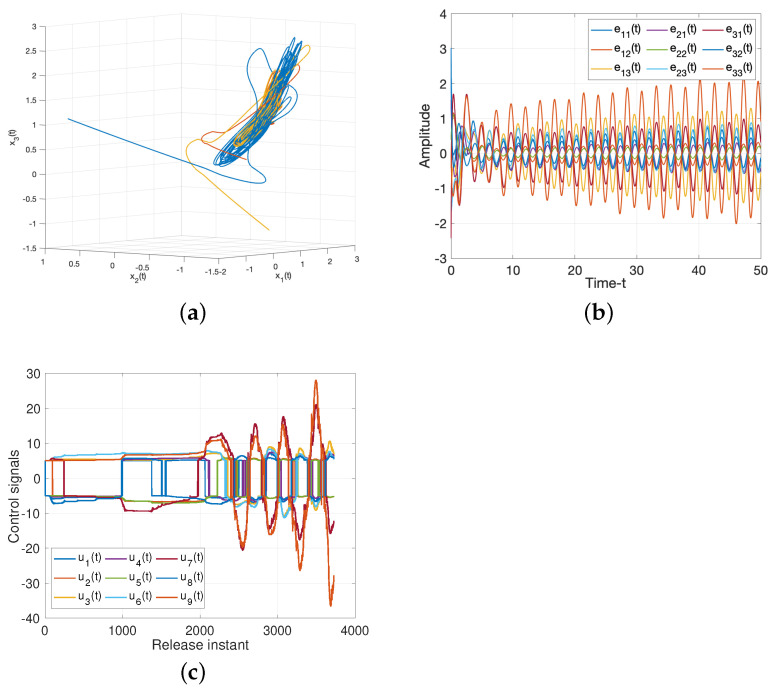
Dynamic trajectories of systems (Equation 11) and (Equation 12) without controller. (**a**) Chaotic sequences of drive system (Equation 11); (**b**) States of error system (Equation 18); (**c**) Input of the ineffective control.

**Figure 4 entropy-25-01241-f004:**
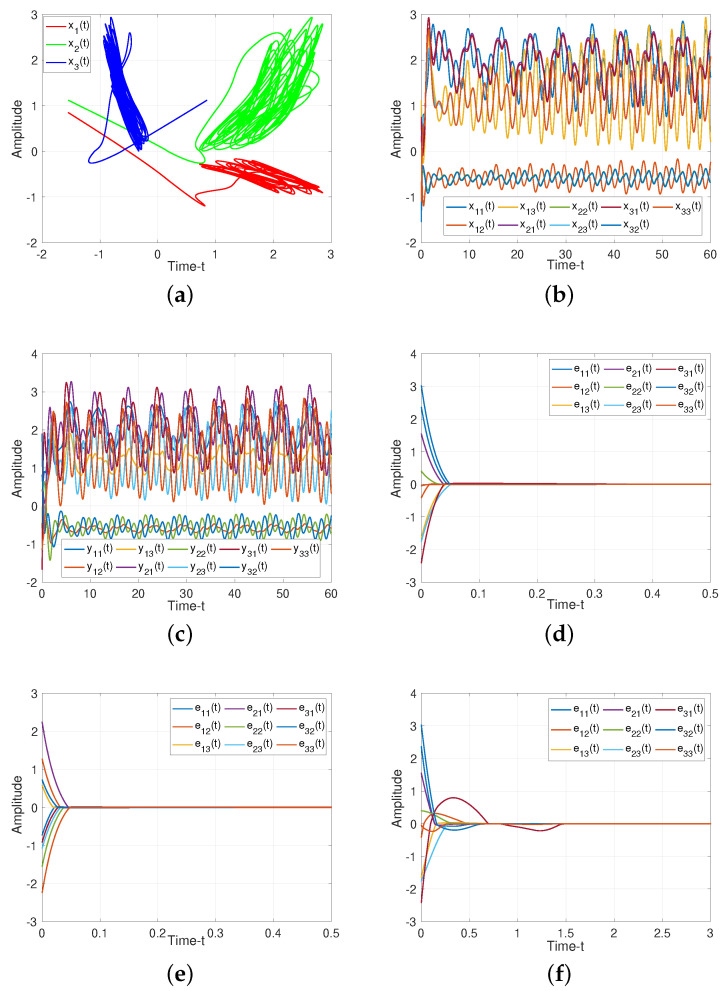
Dynamic trajectories of systems (Equation 11) and (Equation 12). (**a**) Chaotic sequences of drive system (Equation 11); (**b**) States of drive system (Equation 11) without controller; (**c**) States of response system (Equation 12) without controller; (**d**) Synchronization error of system (Equation 18) under feedback controller (Equation 19) with α=1; (**e**) Anti-synchronization error of system(Equation 18) under feedback controller (Equation 19) with α=−1; (**f**) Adaptive-synchronization error of system (Equation 18) under adaptive controller (Equation 41).

**Figure 5 entropy-25-01241-f005:**
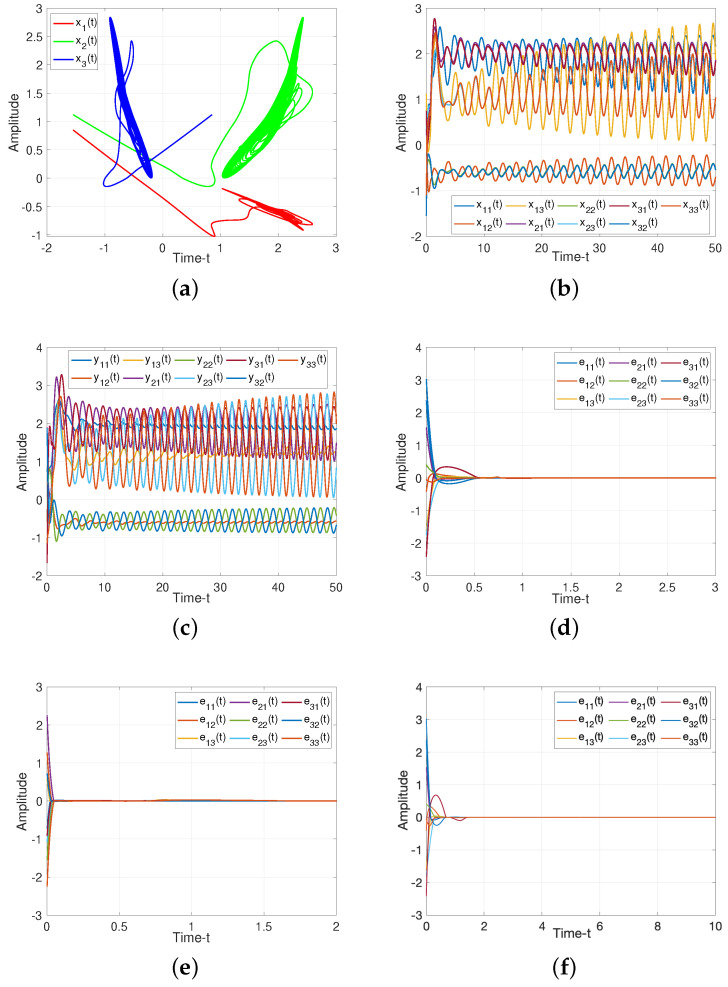
Dynamic trajectories of systems (Equation 11) and (Equation 12) without uncertainties. (**a**) Chaotic sequences of drive system (Equation 11); (**b**) States of drive system (Equation 11) without controller; (**c**) States of response system (Equation 12) without controller; (**d**) Synchronization error of system (Equation 18) under feedback controller (Equation 19) with α=1; (**e**) Anti-synchronization error of system (Equation 18) under feedback controller (Equation 19) with α=−1; (**f**) Adaptive-synchronization error of system (Equation 18) under adaptive controller (Equation 41).

**Figure 6 entropy-25-01241-f006:**
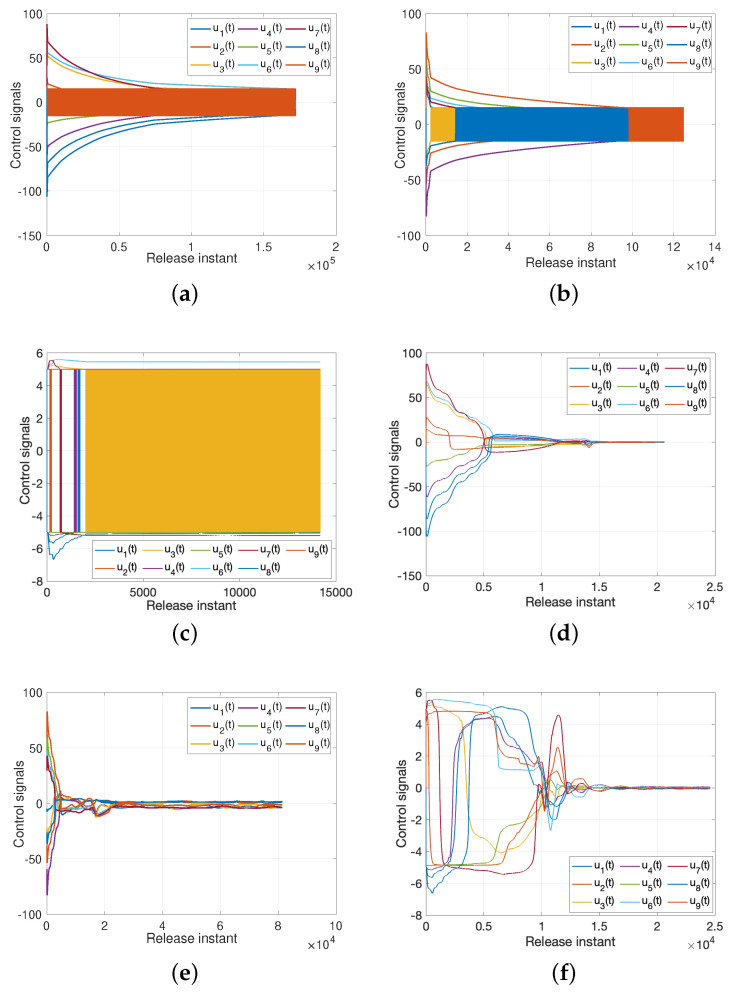
Inputs from different types controllers. (**a**) Feedback controller (Equation 19) with buffeting phenomenon; (**b**) Feedback anti-controller with buffeting phenomenon; (**c**) Adaptive controller (Equation 41) with buffeting phenomenon; (**d**) Feedback controller (Equation 19) without buffeting phenomenon; (**e**) Feedback anti-controller without buffeting phenomenon; (**f**) Adaptive controller (Equation 41) without buffeting phenomenon.

**Figure 7 entropy-25-01241-f007:**
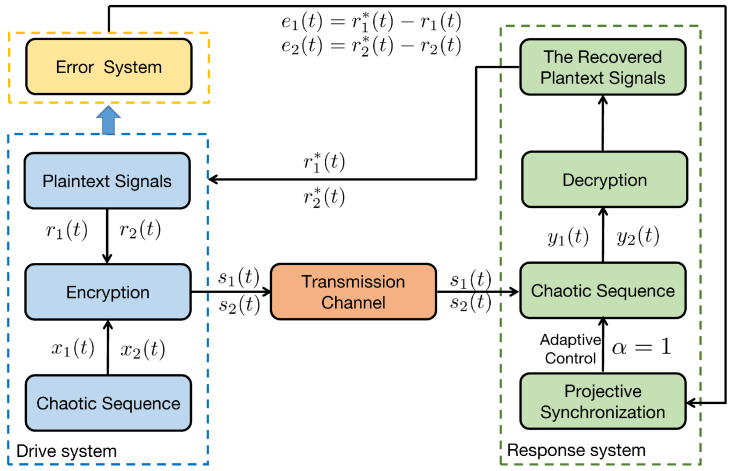
Secure communication process of proposed algorithm.

**Figure 8 entropy-25-01241-f008:**
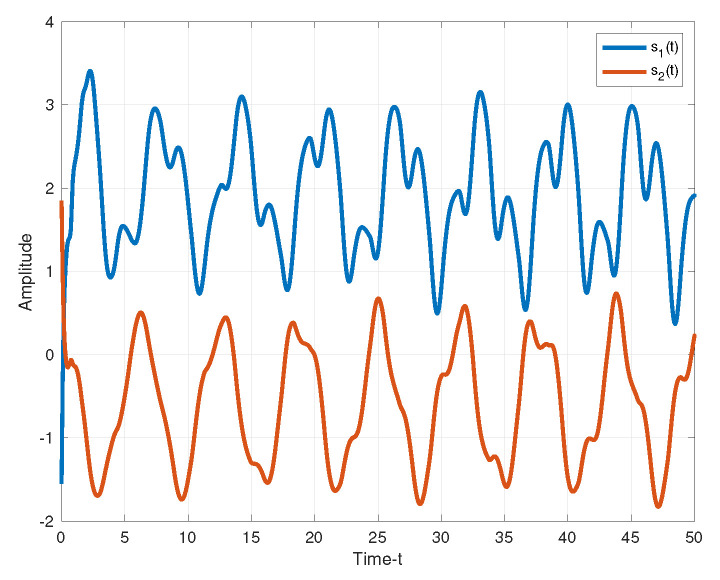
The transmitted signals.

**Figure 9 entropy-25-01241-f009:**
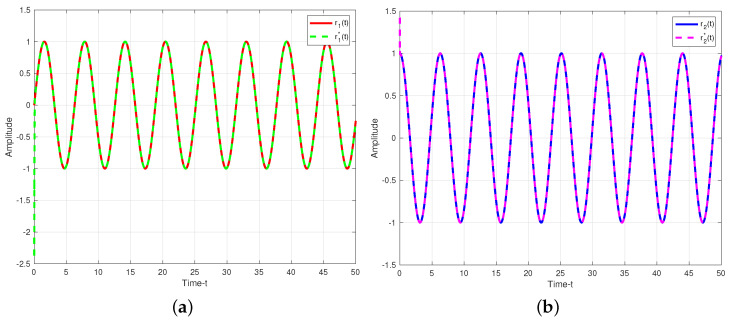
Trajectories of plaintexts and decrypted siganls under adaptive control approach. (**a**) $r_{1}\left(t\right)=\sin\left(t\right)$; (**b**) $r_{2}\left(t\right)=\cos\left(t\right)$.

**Figure 10 entropy-25-01241-f010:**
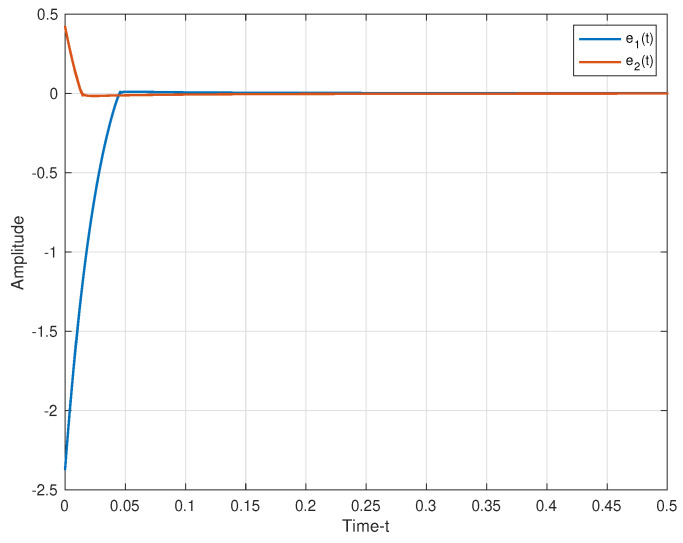
The error between plaintexts and decrypted siganls.

## Data Availability

Not applicable.

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
