# Peer review of "Projective Synchronization of Delayed Uncertain Coupled Memristive Neural Networks and Their Application"

_entropy, 2023, doi:10.3390/e25081241_

Round 1

Reviewer 1 Report

The authors investigated the projective synchronization of coupled multi-links MNNs with uncertainties and delays. Aiming at the issue of parameter mismatch, the principle of extended comparison and a new approach is employed to transform the proposed system into the one with interval parameters. Furthermore, according to the designed LKF, several sucient criteria for projective synchronization are derived under the nonlinear controller. Finally, the chaotic sequences of drive and response systems are applied in signal encryption and decryption of secure communication.  The results provided are meaningful.

Good

Reviewer 2 Report

Dear Authors,

Your work is very well structured and with a strong mathematical consolidation. I ask you only the following questions:

- Has the system defined by Eq.(2) been studied before? If so, please provide the appropriate bibliographic reference. Otherwise, it should be duly clarified to the readers.

- The functions defined by Eqs.(7), (8) and (9) are measurable in what sense?

Finally, I commend the mathematical rigor in the writing and presentation of the results, and the numerical simulations exposed.

Best regards,

R.

Reviewer 3 Report

I have revised carefully this interesting paper. The main problem is that the memristive model is very simple and it is not reflecting the real behavior of solid-state memristors or emulators. Further, the behavioral model of the memristor is used as nonlinear function, but not as memristor. Please give me more details on these issues.

Reviewer 4 Report

The comments are as follows.

1. The projective synchronization is not clear since the parameter \alpha is now defined in (12).

2. Compared with control with discontinuous characteristics, what are the advantages of adaptive control? Besides, there is a sign function in the proposed controller, and buffeting phenomenon will exist.

3. The secure communication is not clear, the encryption image and the gray histogram is not reflected. The related image encryption by using the synchronization of memristive systems can be referred to Finite-/fixed-time synchronization of memristor chaotic systems and image encryption application, synchronization of a memristor chaotic system and image encryption.

4. The example 2 use the chaotic sequences. However, system in example 1 is not a chaotic system. 

5. Lots of mistakes need to be checked and corrected. Please see the comments on the Quality of English Language.

The quality of English can be improved. Please check the whole paper.

1. but the ……will leads to ……

2. The tense must be consistent in Abstract and Conclusion.

3. II, III, IV, V.

4. Following are the novelties of current articles.

5. initial conditions.

6. r_0, r_1 and \tau_0, \tau_1 are not consisten.

7. What R^T means?

8.  There exist real constant.

9. such that, such as

Round 2

Reviewer 3 Report

My questions were clarified. Now the paper has the quality required

Author Response

Dear Reviewer,

Thanks for your message, we have updated the manuscript as reviewer 4 suggested.

Reviewer 4 Report

The response is almost done. It is helpful for enhancing the secure communication by comparing the application part with the aforementioned two application papers and adding the discussions. Besides, the icon size of some images should be enlarged, such as Figs. 3,4,5,6.

Fine

Author Response

Dear Reviewer,

Thanks for your message and suggestion. We have updated the manuscript as you suggested, please let us know if there is anymore comments that you would like to post.
